# Abrupt and spontaneous strategy switches emerge in simple regularised neural networks

**Anika T. Löwe**[1,2,3]*, **Léo Touzo**[4], **Paul S. Muhle-Karbe**[5,6,7], **Andrew M. Saxe**[8,9,10], **Christopher Summerfield**[5‡], **Nicolas W. Schuck**[1,2,3‡]

**1** Max Planck Research Group NeuroCode, Max Planck Institute for Human Development, Berlin, Germany, **2** Max Planck UCL Centre for Computational Psychiatry and Ageing Research, Berlin, Germany, **3** Institute of Psychology, Universität Hamburg, Hamburg, Germany, **4** Laboratoire de Physique de l'Ecole Normale Supérieure, CNRS, ENS, Université PSL, Sorbonne Université, Université Paris Cité, Paris, France, **5** Department of Experimental Psychology, University of Oxford, Oxford, United Kingdom, **6** School of Psychology, University of Birmingham, Birmingham, United Kingdom, **7** Centre for Human Brain Health, University of Birmingham, Birmingham, United Kingdom, **8** Gatsby Computational Neuroscience Unit, University College London, London, United Kingdom, **9** Sainsbury Wellcome Centre, University College London, London, United Kingdom, **10** CIFAR Azrieli Global Scholar, CIFAR, Toronto, Canada

‡ These authors are joint senior authors on this work.
* loewe@mpib-berlin.mpg.de

**Data Availability Statement:** Code is freely available on gitlab in this repo: https://gitlab.com/aloewe/insightnets Data are fully available on OSF:

## Abstract

Humans sometimes have an insight that leads to a sudden and drastic performance improvement on the task they are working on. Sudden strategy adaptations are often linked to insights, considered to be a unique aspect of human cognition tied to complex processes such as creativity or meta-cognitive reasoning. Here, we take a learning perspective and ask whether insight-like behaviour can occur in simple artificial neural networks, even when the models only learn to form input-output associations through gradual gradient descent. We compared learning dynamics in humans and regularised neural networks in a perceptual decision task that included a hidden regularity to solve the task more efficiently. Our results show that only some humans discover this regularity, and that behaviour is marked by a sudden and abrupt strategy switch that reflects an aha-moment. Notably, we find that simple neural networks with a gradual learning rule and a constant learning rate closely mimicked behavioural characteristics of human insight-like switches, exhibiting delay of insight, suddenness and selective occurrence in only some networks. Analyses of network architectures and learning dynamics revealed that insight-like behaviour crucially depended on a regularised gating mechanism and noise added to gradient updates, which allowed the networks to accumulate "silent knowledge" that is initially suppressed by regularised gating. This suggests that insight-like behaviour can arise from gradual learning in simple neural networks, where it reflects the combined influences of noise, gating and regularisation. These results have potential implications for more complex systems, such as the brain, and guide the way for future insight research.

https://osf.io/wh2r8/?view_only=
5c5136bd1deb4383b50a0751a97df5b6.

**Funding:** ATL was supported by the Max-Planck-Gesellschaft (IMPRS COMPPPSYCH). PSMK was funded by the Wellcome Trust (210849/Z/18/Z). AMS was funded by the Wellcome Trust and Royal Society (216386/Z/19/Z), the Wellcome Trust (219627/Z/19/Z) and the Gatsby Charitable Foundation (GAT3755). CS was funded by HORIZON EUROPE European Research Council (Human Brain Project 945539) and the H2020 European Research Council (ERC Consolidator 725937). NWS was funded by the Max-Planck-Gesellschaft (M.TN.A.BILD0004), the European Union (ERC StG-REPLAY-852669) and the BMBF (Excellence Strategy of the Federal Government and the Länder). The funders had no role in study design, data collection, data analysis, data interpretation, or writing of the report.

**Competing interests:** The authors have declared that no competing interests exist.

## Author summary

Insights, or aha-moments, are a remarkable phenomenon in human cognition that is unique in a number of ways: they are accompanied by a powerful subjective experience, occur abruptly after an unpredictable period of having been stuck on a problem, and for some arise never. But are insights harbingers of a unique mode of learning that only appears in the highest of animals? We show that insight-like behaviours can occur even in the simplest neural networks trained with regular gradient descent techniques. Human and machine behaviour was compared on the same decision making task which included a hidden regularity. Neural networks with L1-regularised gate modulation closely mimic the key behavioural characteristics that we identified in humans. An analysis of the insight networks showed that noise and regularisation played an important part in bringing about insight-like behaviour, besides being preceded by "silent knowledge" that is initially suppressed by gating. Our results shed new light on the computational origins of insights and suggest that they can arise from gradual learning mechanisms.

## Introduction

Neural networks trained with stochastic gradient descent (SGD) are a current theory of human learning that can account for a wide range of learning phenomena. At face value, SGD trained network models seem to imply that all learning is gradual. Yet, humans sometimes learn in an abrupt manner and improve on task in a seemingly spontaneous way. These striking cases have been related to insights or aha-moments [1, 2], which are thought to reflect a qualitatively different, discrete learning mechanism [3, 4]. One prominent idea, dating back to Gestalt psychology [1, 5], is that an insight occurs when an agent has found a novel problem solution by restructuring an existing task representation [6, 7]. It has also been noted that humans often lack the ability to trace back the cognitive process leading up to an insight [8], suggesting that insights involve unconscious processes becoming conscious. Moreover, so called "aha-moments" can sometimes even be accompanied by a feeling of relief or pleasure in humans [6, 9, 10]. Such putative uniqueness of the insight phenomenon would also be in line with work that has related insights to brain regions distinct from those associated with gradual learning [8, 11, 12]. Altogether, these findings have led psychologists and neuroscientists to propose that insights are governed by a distinct learning or reasoning process [8] that cannot be accounted for by common gradual theories of learning.

   Here, we show that sudden and abrupt changes in behaviour in a task that elicits insights in humans can occur in a simple gradual learning system devoid of any dedicated insight mechanism. Our argument does not concern the subjective experiences related to insights, but focuses on showing how insight-like *behaviour* can emerge from gradual learning algorithms. Specifically, we aim to explain the following three behavioural observations [13–17]: First, insights trigger abrupt behavioural changes. These sudden behavioural changes are often accompanied by fast neural transitions [2, 13, 18–20], and by meta-cognitive suddenness (a "sudden and unexpected flash") [4, 16, 21–23]. Second, insights occur selectively in some subjects, while for others improvement in task performance arises only gradually, or never [13]. Finally, insights occur "spontaneously", i.e. without the help of external cues [24], and are therefore observed after a seemingly random duration of impasse or delay [7] that differs between participants. In other words, participants seem to be "blind" to the new solution for

an extended period of time, before it suddenly occurs to them. Insights are thus characterised by a suddenness, selectivity, and variable delay of occurrence.

Current computational accounts of insight have proposed a number of specific mechanisms that operate in parallel to gradual learning and cause abrupt strategy changes linked to aha-moments. One interesting model that can dynamically switch between strategies was reported by Collins and colleagues [25, 26]. The model is able to maintain multiple concurrent behavioural strategies, switch between them based on current task requirements, and devise new strategies. Other models have focused on representational restructuring [27], integration of explicit and implicit knowledge [28] or metacognitive monitoring [29]. Our work introduces a much simpler but unified model where both gradual and insight-like learning stem from a singular, delta-rule-based algorithm. Our model distinguishes itself from previous approaches by utilising this single updating rule before, during and after strategy switches, eliminating the need for dedicated strategy monitoring, maintenance, or multiple memory systems. As we will show below, our model also does not require a heightened occurrence of errors or low rewards to switch strategy. To emphasise our theoretical point, we focus on the most concise model that can exhibit insight-like behaviour. This is achieved through a simple neural network comprising merely two input nodes and one output node, whereby each input node is regulated by a single multiplicative gate that modulates its respective weight on the output. During learning, gates are L1-regularised and noise is added to the gradients. Stripped down to such minimal assumptions, our model demonstrates how insight-like behaviour can theoretically emerge from a system devoid of complex mechanisms such as restructuring. Furthermore, we show how our model qualitatively and quantitatively aligns with human behaviour across the three insight dimensions suddenness, selectivity and delay.

The idea that insight-like behaviour can arise from gradual learning is supported by previous work on human behaviour [30] and neural networks trained with gradient descent [31]. Saxe and colleagues [32], for instance, have shown that non-linear learning dynamics, i.e. suddenness in the form of saddle points and stage-like transitions, can result from gradient descent even in linear neural networks, which could explain sudden behavioural improvements. Other work has shown a delayed or stage-like mode of learning in neural networks that is reminiscent of the period of impasse observed in humans, reflecting for instance the structure of the input data [33–35], or information compression of features that at some point seemed task-irrelevant [36, 37]. Finally, previous work has also found substantial individual differences between neural network instances that are induced by random differences in weight initialisation, noise, or the order of training examples [38, 39], which can become larger with training [40]. Notably, while different behavioural aspects of sudden and abrupt strategy switches have been shown, so far no study has made a detailed comparison to behaviour in humans and specifically asked whether delay, suddenness and selectivity can occur jointly in a single network model.

Two factors that influence discontinuities in learning in neural networks are regularisation and gating. A simple mechanism to attain regularisation involves adding a penalty term to the error function that prevents coefficients from reaching large values, and which thereby leads to suppression of input features [41]. While these forms of explicit regularisation share similarities with other (implicit) regularisation techniques, these two forms are not identical and we focus only on the former. From a cognitive neuroscience perspective, regularisation may correspond to mechanisms that limit the number of factors that are taken into account during decision making, reminiscent of the effects of priors or attentional mechanisms on human cognition [42]. Crucially, while regularisation is useful in that it avoids overfitting or getting stuck in a local minimum [43], the lingering suppression of some inputs might also cause above-mentioned "blindness" to a solution. The second factor—gating—describes a

multiplicative interaction between learnable parameters and is ubiquitously found in real neurons whose activity is often gain modulated [44]. It is known that such multiplicative interactions cause exponential transitions in learning, which are for instance widely used in multiplicative dynamical systems like the logistic growth model or recurrent neural networks. Hence, regularisation and gating are both are commonly used in artificial neural networks [41, 45, 46], and are inspired by biological brains [47–49]. This makes regularisation and gating natural candidate aspects of network structure and training that could be related to insight-like behaviour, as evidenced by a temporary impasse followed by a sudden performance change.

A simple neural network architecture with multiplicative gates and L1-regularisation served as our candidate model. We focused specifically on L1-regularisation—which penalises the loss by the absolute rather than quadratic gate values—as it forces gates of irrelevant inputs most strongly towards 0, causing a sustained suppression period before the fast transition, similar to the impasse observed in humans. Our model is meant to be conceptual and not biologically realistic, as we aim to demonstrate how insight-like behaviour and fast representational restructuring can occur in simple architectures. We also show that our findings generalise to more complex neural networks with multiple input nodes and hidden layers which show qualitatively similar behaviour to what is described below.

We study insight-like strategy switches using a task where participants initially learn a functional, but suboptimal, strategy. This strategy is spontaneously replaced by some participants with a more optimal solution, mirroring an insight-like process [13–15, 20]. Participants are not made aware of a superior strategy. Our task therefore differs from other insight tasks where participants are asked to actively search for a novel problem solution (e.g. Remotes Associates Tasks [50] or Compounds Remotes Associates Tasks [51]). Nevertheless, the task aligns with the core concept of almost all insight tasks, which require a modification of the initial problem representation [23]. Indeed, our task is very similar to the well established Number Reduction Task [52–54], and most suitable to formal analysis of insight since participants' knowledge can be tracked with high temporal resolution [55].

## Results

To study insight-like learning dynamics, 99 participants and 99 neural networks performed the Spontaneous Strategy Switch Task [13, 15], which required a binary choice about a stimulus characterised by two features. A circular array of coloured and moving dots [56] served as the stimulus in humans (Fig 1A and 1C), while two scalar inputs represented the stimulus features symbolically for networks (Fig 1D). In humans, the motion coherence was varied in five levels such as to produce an accuracy gradient, while the dot colour was easy to recognise throughout (either orange or purple, see below). The network inputs for motion were set such that for each human one network performed with the same behavioural accuracy in the different coherence levels (see below for details). Participants and networks had to learn the correct choice in response to each stimulus from trial-wise binary feedback (Fig 1C), and were not instructed which features of the stimulus to pay attention to.

Participants first underwent an initial *training phase* with high motion coherence (2 blocks, 100 trials each in humans; 6 blocks in networks), followed by a *motion phase* that marked the onset of motion coherence variability (2 blocks in humans and networks). During both phases, only the motion direction predicted the correct choice, while stimulus colour was random (see Fig 1B). Without any announcement, stimulus colour became predictive of the correct response in the *motion and colour phase*, such that from then on both features could be used to determine choice (5 blocks for humans and networks, Fig 1B). The experiment concluded

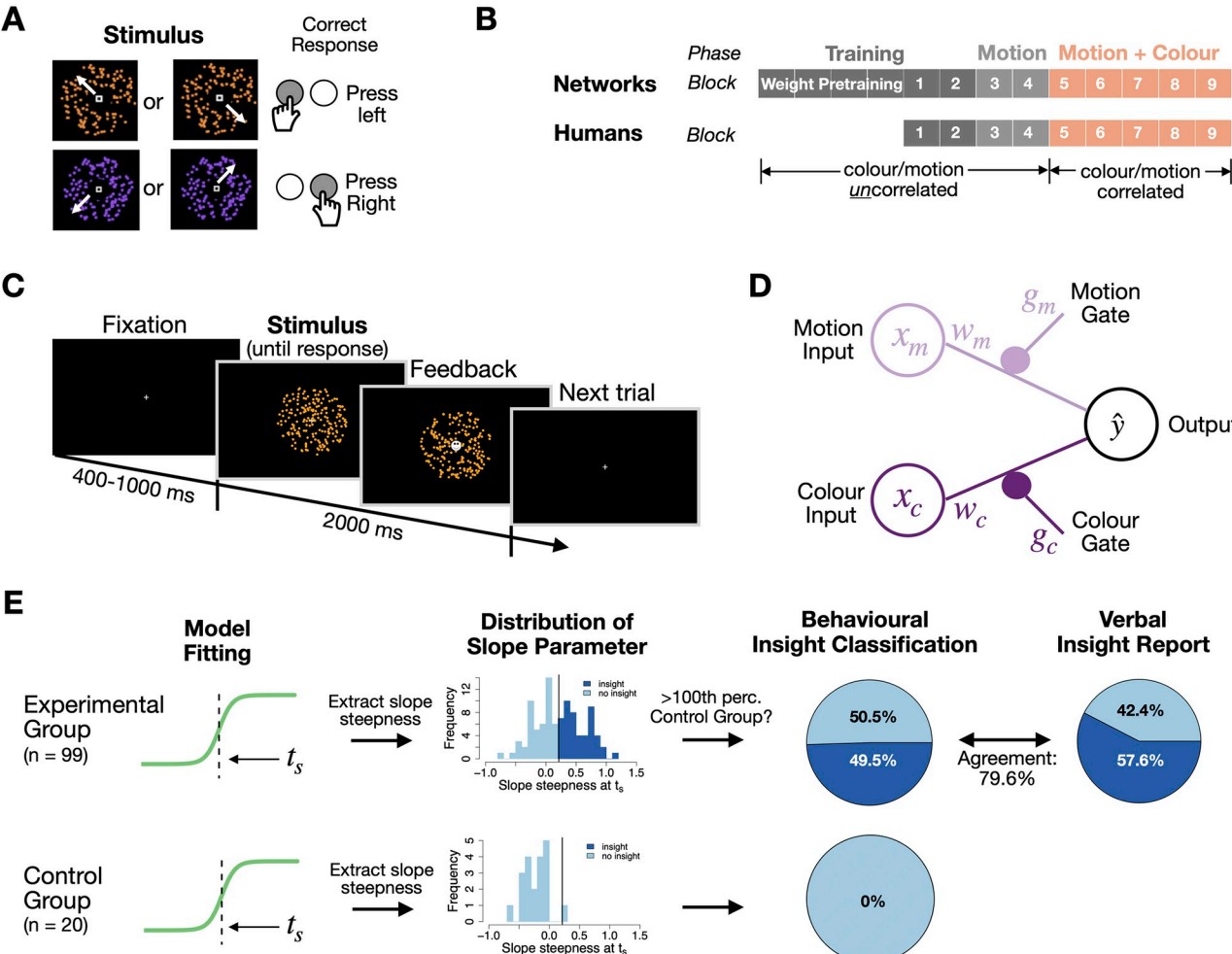

**Fig 1.** Stimuli, task design and insight classification procedure **(A)** Stimuli and stimulus-response mapping: dot clouds were either coloured in orange or purple and moved to one of the four directions NW, NE, SE, SW with varying coherence. A left response key, "X", corresponded to the NW/SE motion directions, while a right response key "M" corresponded to NE/SW directions. **(B)** Task structure of the two-alternative forced choice task for humans and neural networks: each block consisted of 100 trials. A first training block only for humans contained only 100% motion coherence trials to familiarise subjects with the S-R mapping. The remaining training blocks contained only high coherence (0.2, 0.3, 0.45) trials. In the *motion phase*, colour changed randomly and was not predictive and all motion coherence levels were included. Colour was predictive of correct choices and correlated with motion directions as well as correct response buttons only in the last five blocks (*motion and colour phase*). Participants were instructed to use colour before the very last block 9, which served as sanity check (data shown only in SI). **(C)** Trial structure: a fixation cue is shown for a duration that is shuffled between 400, 600, 800 and 1000 ms. The random dot cloud stimulus is displayed for 2000 ms. A response can be made during these entire 2000 ms, but a central feedback cue will replace the fixation cue immediately after a response. **(D)** Schematic of the neural network with regularised gate modulation used to model insights. **(E)** Insight classification procedure: We fitted a sigmoid model ($y = \frac{y_{max} - y_{min}}{1 + e^{-m(t - t_s)}} + y_{min}$, see Methods for details) to data from the lowest motion coherence condition data of both the Experimental Group and a Control Group (where colour never becomes predictive), and derived the distributions of the slope steepness at the estimated switch point (inflection point $t_s$ of sigmoid function). We then asked which participants from the Experimental group had fitted slopes that were steeper than the 100th percentile of the Control Group. The resulting purely behavioural classification of insights agreed to 79.6% with verbal insight reports from a post-task questionnaire and predicted a number of behavioural features (see text). Importantly, using this method allowed us to apply the same procedure to neural networks.

with a questionnaire, in which participants were asked whether (1) they had noticed a rule, (2) how long it took them to notice it, (3) whether they had paid attention to colour during choice. The questionnaire was followed by an instruction block that served as a sanity check (see Methods).

Phases were not cued and or announced to participants. Previous work has shown that participants discover the hidden opportunity to use the stimulus colour that arises in the *motion and colour phase* through insight, as evidenced by sudden, delayed and selective behavioural changes that go hand in hand with gaining consciousness about the new regularity [15]. This setup differs from traditional problem-solving tasks where participants actively seek solutions [8, 9, 57], but it closely aligns with the often used Number Reduction Task (NRT) [54], where abrupt task improvements emerge through insights about uninstructed task elements, even though the task can in principle be performed using the initially learned/instructed strategy. Note that pretraining and low coherence trials were only included because they facilitate analysis and the comparability to neural networks; the existence of these aspects alone is not necessary for insights to occur (see [15] and below for control experiments).

## Human behaviour

Our results are in line with multiple previous reports demonstrating the below reported switches and their properties in humans [13, 15, 20].

**Baseline task performance (Training and Motion Phases).**   Data from the *training phase*, during which motion directions were highly coherent but uncorrelated to colours (Block 1–2, dark grey tiles in Fig 1B), showed that participants learned the response mapping for the four motion directions well (78% correct, t-test against chance: $t(98) = 30.8$, $p < .001$). In the following *motion phase*, noise was added to the motion, while the colour remained uncorrelated (blocks 3–4, grey tiles in Fig 1B). This resulted in an accuracy gradient that depended on noise level (linear mixed effects model of accuracy: $\chi^2(1) = 726.36$, $p < .001$; RTs: $\chi^2(1) = 365.07$, $p < .001$; N = 99, Fig 2A). Crucially, while performance in the condition with least noise was very high (91%), it was heavily diminished in the conditions with the largest amounts of motion noise, i.e. the two lowest coherence conditions, where participants only performed at only 60% and 63%, and did not change over time (paired t-test block 3 vs 4: $t(195.9) = -1.13$, $p = 0.3$, $d = 0.16$). Hence, substantial performance (improvements) in the high noise condition can only be attributed to colour use, rather than heightened motion sensitivity.

**Task performance after correlation onset (Motion and Colour Phase).**   The noise level continued to influence performance in the *motion and colour phase*, as evidenced by a difference between performance in high vs. low coherence trials (20, 30 & 45% vs 5 & 10% coherent motion, respectively; $M = 93 \pm 6\%$ vs $M = 77 \pm 12\%$; $t(140.9) = 12.5$, $p < .001$, $d = 1.78$, see Fig 2A and 2B). Notably, however, the onset of the colour correlation triggered performance improvements across all coherence levels ($t(187.2) = -12.4$, $p < .001$, $d = 1.8$; end of *motion phase*: $M = 78 \pm 7\%$ vs. end of *motion and colour phase*: $M = 91 \pm 8\%$), contrasting the stable performance found during the motion phase and suggesting that at least some participants leveraged colour information once available.

**Insight classification.**   We asked whether these improvements are related to gaining conscious insight by analysing the post-experimental questionnaire (Fig 1E, right). Results show that conscious knowledge about the colour regularity arose in some, but not all, participants: 57.6% (57/99) reported in the questionnaire to have used colour, while 42.4% indicated to not have noticed or used the colour. Hence insights were selective. As expected, the verbal report also related to performance differences in the lowest coherence condition in Block 8, where participants performed at 91.4% vs 68.7% if they had reported an insight vs not.

Since neural networks cannot provide questionnaire answers, we developed a behavioural classification of insight-like strategy switches. Previous work [13–15] has used a simple performance threshold of 75% in low coherence trials that also correlates highly with verbal insight reports in our sample (see above, $r(97) = .67$, $p < .001$). But this metric is not specifically

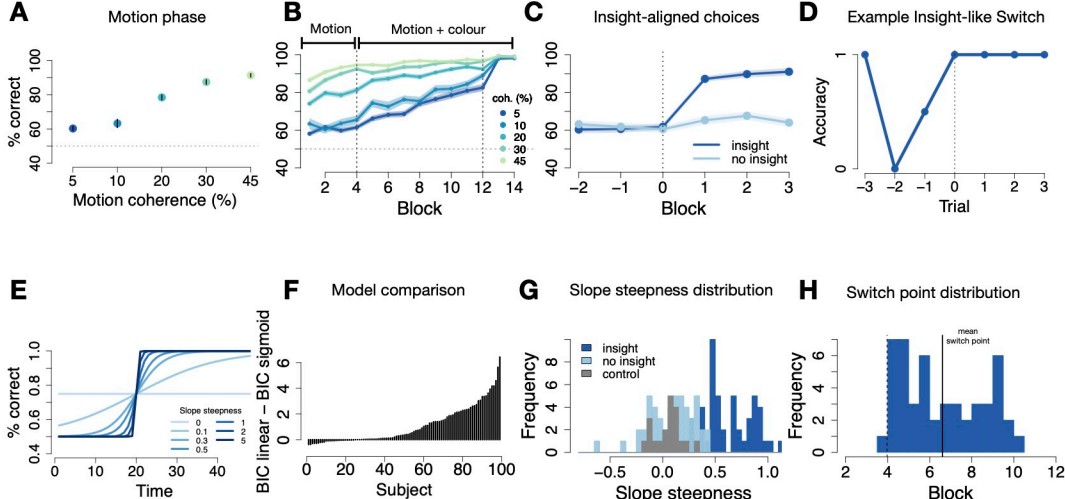

**Fig 2.** Humans: task performance and insight-like strategy switches **(A)** Accuracy (% correct) during the *motion phase* increases with increasing motion coherence. N = 99, error bars signify standard error of the mean (SEM). **(B)** Accuracy (% correct) over the course of the experiment for all motion coherence levels. First dashed vertical line marks the onset of the colour predictiveness (*motion and colour phase*), second dashed vertical line the "instruction" about colour predictiveness. Blocks shown are halved task blocks (50 trials each). **(C)** Switch point-aligned accuracy on lowest motion coherence level for insight (49/99) and no-insight (50/99) subjects. Blocks shown are halved task blocks (50 trials each). N = 99, error shadows signify SEM. **(D)** Trial-wise switch-aligned smoothed binary responses on lowest motion coherence level for an example insight subject. **(E)** Illustration of the sigmoid function for different slope steepness parameters. **(F)** Difference between BICs of the linear and sigmoid function for each human subject. N = 99. **(G)** Distributions of fitted slope steepness at inflection point parameter for control experiment and classified insight and no-insight groups. **(H)** Distribution of switch points. Dashed vertical line marks onset of colour predictiveness. Blocks shown are halved task blocks (50 trials each).

related to suddenness, and therefore might identify participants who learned gradually about the colour. We therefore used a cross-validated approach in which we first identified the maximum suddenness than can occur by chance in a control experiment, and used this threshold for our main sample (Fig 1E, the same procedure was applied to networks). Details about the control experiment, in which participants (N = 20) performed an identical task, except that colour never started to correlate with motion, and hence no insight was possible, can be found in the Methods.

To calculate suddenness, we first fitted a simple sigmoid function to participant accuracy with free parameters, slope $m$, inflection point $t_s$ and function maximum $y_{max}$ (details see Methods). We then calculated the rate of performance change at the inflection point (see Methods) of the fitted sigmoid function, and asked how many subjects from the experimental group had steeper behavioural transitions than the maximum value observed in the control group. This showed that about half of participants (49/99, 49.5%) had values larger than the 100% percentile of the control distribution (Fig 1E). Hence, in some participants behaviour changed so suddenly as to suggest insights (Fig 2F). While this behavioural classification resulted in a more conservative estimate of the number of insight participants compared to the questionnaire, it highly overlapped with these verbal reports. Of the 49 participants classified as insight subjects 39 (79.6%) also self-reported to have used colour to make correct choices, see Figs 1E, and JA and JB in S1 Text. This again correlates highly with the previously used performance threshold ($r(97) = .44$, $p < .001$). Hence, our behavioural marker of unexpectedly sudden performance changes can serve as a valid indicator for sudden insight.

Note that while in the above model fitting we used bins of 15 trials to identify suddenness, behavioural transitions often occurred within just a few trials (Fig G in S1 Text), i.e. on the order of 15–30 seconds, as is common for insights. The averaging was necessary in order to stabilise model fits. We also note that our choice of fitting function more generally was validated by group-wise model comparisons against a linear ramp (free parameters intercept $y_0$ and slope $m$) or a step function (free parameters inflection point $t_s$ and function max $y_{max}$), BICs -6.7, -6.4 and -6.5, protected exceedance probabilities: 1, 0, 0, for sigmoid, linear and ramp models, respectively (see Figs 2D, 2E and B in S1 Text).

**Behavioural differences between participants with and without insights.** We validated our behavioural metric of selectivity through additional analyses. Splitting participants into separate insight and no-insight groups based on the above procedure showed that, as expected based on the dependency of accuracy and our behavioural metric, insight subjects started to perform significantly better in the lowest coherence trials once the *motion and colour phase* (Fig 2C) started, (mean proportion correct in *motion and colour phase*: $M = 83 \pm 10\%$), compared to participants without insight ($M = 66 \pm 8\%$) ($t(92) = 9.5$, $p < .001$, $d = 1.9$). Unsurprisingly, a difference in behavioural accuracy between insight participants and no-insight participants also held when the average across all coherence levels was considered ($M = 91 \pm 5\%$ vs. $M = 83 \pm 7\%$, respectively, t-test: $t(95.4) = 6.9$, $p < .001$, $d = 1.4$). Accuracy in the *motion phase*, which was not used in steepness fitting, did not differ between groups (low coherence trials: $M = 59\%$, vs. $M = 62\%$; $t(94.4) = -1.9$, $p = 0.07$, $d = 0.38$; all noise levels: $M = 76\%$ vs $M = 76\%$, $t(96) = 0.45$, $p = 0.7$, $d = 0.09$). Reaction times, which are independent from the choices used in model fitting and thus served as a sanity check for our behavioural metric split, reflected the same improvements upon switching to the colour strategy. Subjects who showed insight about the colour rule ($M = 748.47 \pm 171.1$ ms) were significantly faster ($t(96.9) = -4.9$, $p < .001$, $d = 0.97$) than subjects that did not ($M = 924.2 \pm 188.9$ ms) on low coherence trials, as well as over all noise levels ($t(97) = -3.8$, $p < .001$, $d = 0.87$) ($M = 675.7 \pm 133$ ms and $M = 798.7 \pm 150.3$ ms, respectively).

**Delay of insights.** Having established that behavioural changes were sudden and occurred for only some participants, we next asked whether insights occurred with random delays. To quantify this key characteristic, insight moments were defined as the time points of inflection of the fitted sigmoid function, i.e. when performance exhibited abrupt increases (see Methods). We verified the precision of our switch point identification by time-locking the data to the individually fitted switch points. This showed that accuracy steeply increased between the trial bins (50 trials) immediately before vs. after the switch, as expected ($M = 62\%$ vs $M = 83\%$ $t(89) = -11.2$, $p < .001$, $d = 2.34$, Fig 2C and Fig EA in S1 Text). Additionally, reaction times dropped steeply from pre- to post-switch ($M = 971.63$ ms vs. $M = 818.77$ ms, $t(87) = 3.34$, $p < .001$, $d = 0.7$). The average delay of insight onset was 130 trials (±95 trials), corresponding to 2.6 trial bins (Fig 2G). The distribution of delays among insight participants ranged from 0 to 6 trial bins after the start of the *motion and colour phase*, and statistically did not differ from a uniform distribution taking into account the hazard rate (Exact two-sided Kolmogorov-Smirnov test: $D(49) = 0.25$, $p = 0.11$).

Hence, the behaviour of human subjects showed all characteristics of insight: sudden improvements in performance that occurred only in a subgroup and with variable delays.

**Effects of feedback and low coherence.** Two possible objections to the idea that the above described insight-like strategy switches are internally generated and truly spontaneous could be made: first, the non-random nature of motion and feedback provided in the lowest coherence condition could mean that performance on these trials reflects heightened motion sensitivity coupled to reward guided choices, rather than the use of a colour strategy. Second, the presence of low coherence trials in and of itself might prompt participants to search for an

alternative strategy. We dismiss these objections through two control experiments. The first (N = 61) replicated the original task, but incorporated truly random stimuli in the lowest coherence condition (0% coherence) and replaced feedback with instructions. The second (N = 29) introduced the two lowest motion coherence levels (5% and 10%) only in the sixth task block. As detailed in the SI, both experiments elicited a substantial number of insight-like switches, thus supporting the generality of the here reported phenomenon across a number of task variations (in line with Gaschler et al [15]).

## Neural network models

To probe whether insight-like behaviour can arise in simple neural networks trained with gradient descent, we simulated 99 network models performing the same decision making task.

**Architecture.** The networks had two input nodes ($x_c$, $x_m$, for colour and motion, respectively), two input-specific gates ($g_m$, $g_c$) and weights ($w_m$, $w_c$), and one output node ($\hat{y}$, Fig 1D). Network weights and their respective gates were initialised at 0.01. Gates only differ from the weights by the applied regularisation. The network multiplied each input node by two parameters, a corresponding weight, and a gate, and returned a decision based on the output node's sign $\hat{y}$:

$$\hat{y} = \text{sign}(g_m w_m x_m + g_c w_c x_c + \eta) \tag{1}$$

where $\eta \sim \mathcal{N}(0, \sigma = 0.05)$ is Gaussian noise, and weights and gates are the parameters learned online through gradient descent. We note that our architecture is functionally equivalent to a 2-layer diagonal linear network with L1-regularisation on the second layer. Hence, although we focus on the gating interpretation, our results can equally be interpreted as pertaining to a particular simple form of linear 2-layer neural networks.

**Learning algorithm.** To train L1-networks we used a simple squared loss function with L1-regularisation of gate weights:

$$\mathcal{L} = \frac{1}{2}(g_m w_m x_m + g_c w_c x_c + \eta - y)^2 + \lambda(|g_m| + |g_c|) \tag{2}$$

with a fixed level of regularisation $\lambda = 0.07$ and a fixed learning rate of $\alpha = 0.6$. $\mathcal{L}$ was minimised by updating weights and gates after trial, while adding Gaussian noise $\xi \sim \mathcal{N}(\mu_\xi = 0, \sigma_\xi = 0.05)$ to each gradient update to mimic learning noise and induce variability between individual networks (same gradient noise level for all networks). This yielded the following trialwise update equations for noisy SGD of the network's weights given the correct decision $y$:

$$\Delta w_m = -\alpha x_m g_m (x_m g_m w_m + x_c g_c w_c + \eta - y) + \xi_{w_m}, \tag{3}$$

and gates,

$$\Delta g_m = -\alpha x_m w_m (x_m g_m w_m + x_c g_c w_c + \eta - y) - \alpha\lambda\,\text{sign}(g_m) + \xi_{g_m} \tag{4}$$

where we have not notated the dependence of all quantities on trial index $t$ for clarity; and analogous equations hold for colour weights and gates with all noise factors $\xi_{g_m}$, $\xi_{w_m}$ etc, following the same distribution.

We used this setup, L1-regularisation of multiplicative gate weights, for two reasons: First, by adding a penalty term to the error function, regularisation leads to a suppression of (irrelevant) input features, which we reasoned would introduce competitive dynamics between the input channels. This competition can lead to non-linear learning dynamics. We used L1-rather than L2 regularisation because it adds an absolute rather than squared penalty that forces gates of irrelevant inputs most strongly towards 0, compared to L2-regularisation,

which is less aggressive in particular once gates are already very small (which we also show empirically in the S1 Text, Figs A, H, I and J in S1 Text). Second, we implemented multiplicative weights and gates because they induce non-linear quadratic and cubic gradient dynamics. Applying L1-regularisation to the gates then will lead to a sustained suppression period before the fast transition (see Methods for details).

**Network training.** The stimulus features motion and colour were reduced to one input node each, which encoded colour/motion direction of each trial by taking on either a positive or a negative value. More precisely, given the correct decision $y = \pm 1$, the activities of the input nodes were sampled from i.i.d. normal distributions with means $\pm M_m$ and $\pm M_c$ and standard deviations $\sigma_m = 0.01$ and $\sigma_c = 0.01$ for colour and motion respectively. While $\sigma$ models perceptual noise, signal to noise ratio, and therefore the difficulty, is determined by the ratio of the means $M_m$ and $M_c$ to sigma. Hence, different difficulty levels in the motion inputs could have equivalently been modelled as changing variances in the presence of a constant mean. In order to match human performance, we fixed the colour mean shift $M_c$ to 0.22, while the mean shifts of the motion node differed by noise level and were fitted individually such that each human participant had one matched network with comparable pre-insight task accuracy in each motion noise condition (see below).

Networks received an extended pre-task training phase of 6 blocks, but then underwent a training curriculum precisely matched to the human task (2 blocks of 100 trials in the *motion phase* and 5 blocks in the *motion and colour phase*, see Fig 1D). We adjusted direction specificity of motion inputs (i.e. difference in distribution means from which $x_m$ was drawn for left vs right trials) separately for each participant and coherence condition, such that performance in the motion phase was equated between each pair of human and network (Fig 3A, see Methods). Moreover, the colour and motion input sequences used for network training were sampled from the same ten input sequences that humans were exposed to. The learning rate of $\alpha = 0.6$ (same for all participants) was selected to match average learning speed.

**Deep neural networks.** We also trained a simple deep neural network on the same task. Briefly, this network also had 2 inputs, $x_m$ and $x_c$, 48 units in a hidden layer, and two outputs $\hat{y}$. The activation function was ReLu and each weight connecting the inputs with a hidden unit had one associated multiplicative gate $g$, where we again applied L1-regularisation on the gate weights $g$. The network was trained on the Cross Entropy loss using stochastic gradient descent with $\lambda = 0.002$ and $\alpha = 0.1$. As for the one-layer network, we trained this network on a curriculum precisely matched to the human task, and adjusted hyperparameters (noise levels) as described above, such that baseline network performance and learning speed were carefully equated between humans and simple deep neural networks as well.

## Behaviour of L1-regularised neural networks

**Overall network performance.** Networks learned the motion direction-response mapping well in the training phase, during which colour inputs changed randomly and output should therefore depend only on motion inputs (75% correct, t-test against chance: $t(98) = 33.1$, $p < .001$, the accuracy of humans in this phase was $M = 76 \pm 6\%$). As in humans, adding noise to the motion inputs (*motion phase*) resulted in an accuracy gradient that depended on noise level (linear mixed effects model of accuracy: $\chi^2(1) = 165.61$, $p < .001$; N = 99, Fig 3A), as expected given that input distributions were set such that network performance would equate to human accuracy (Fig 3A and 3B). Networks also exhibited low and relatively stable performance levels in the two lowest coherence conditions (58% and 60%, paired t-test to assess stability in the *motion phase*: $t(98) = -0.7$, $p = 0.49$, $d = 0.02$), and had a large performance difference between high vs low coherence trials ($M = 88\% \pm 6\%$ vs. $M = 74 \pm 13\%$, $t(137.3) =$

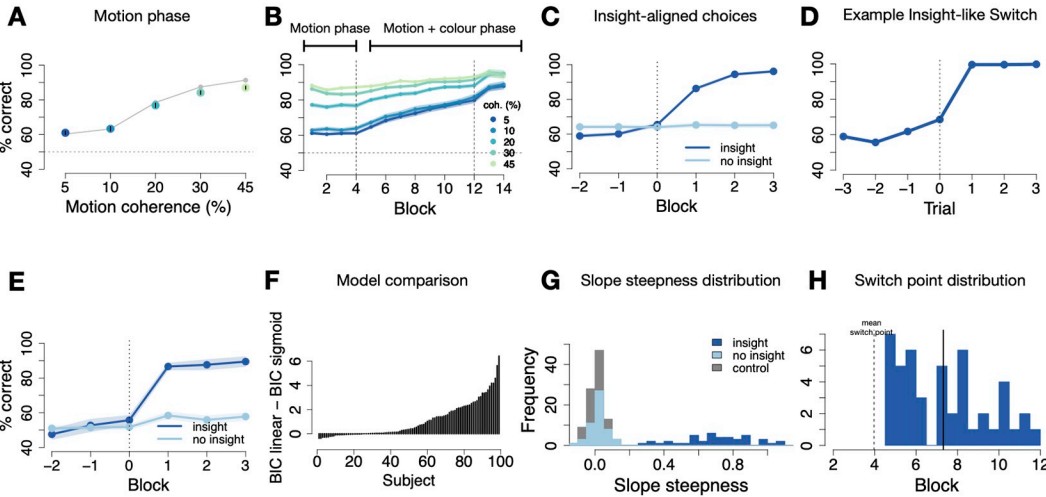

**Fig 3.** L1-regularised neural networks: task performance and insight-like strategy switches **(A)** Accuracy (% correct) during the *motion phase* increases with increasing motion coherence. N = 99, error bars signify SEM. Grey line is human data for comparison. **(B)** Accuracy (% correct) over the course of the experiment for all motion coherence levels. First dashed vertical line marks the onset of the colour predictiveness (*motion and colour phase*), second dashed vertical line the "instruction" about colour predictiveness. Blocks shown are halved task blocks (50 trials each). N = 99, error shadows signify SEM. **(C)** Switch point-aligned accuracy on lowest motion coherence level for insight (46/99) and no-insight (53/99) networks. Blocks shown are halved task blocks (50 trials each). Error shadow signifies SEM. **(D)** Trial-wise switch-aligned continuous outputs on lowest motion coherence level for an example insight network. **(E)** Switch point-aligned accuracy on lowest motion coherence level for insight (18/99) and no-insight (81/99) hidden layer networks. Blocks shown are halved task blocks (50 trials each). Error shadow signifies SEM. **(F)** Difference between BICs of the linear model and sigmoid function for each network. **(G)** Distributions of fitted slope steepness at inflection point parameter for control networks and classified insight and no-insight groups. **(H)** Distribution of switch points. Dashed vertical line marks onset of colour predictiveness. Blocks shown are halved task blocks (50 trials each).

9.6, $p < .001$, $d = 1.36$ for high, i.e. $\geq 20\%$ coherence, vs. low trials). Finally, humans and networks also performed comparably well at the end of learning (last block of the *colour and motion phase*: $M(nets) = 79\% \pm 17\%$ vs. $M(humans) = 82 \pm 17\%$, $t(195.8) = 1.1$, $p = 0.27$, $d = 0.16$, Fig JC in S1 Text), suggesting that at least some networks did start to use colour inputs. Hence, networks' baseline performance and learning were successfully matched to humans.

**Insight-like behavioural characteristics of network behaviour.** To look for characteristics of insight in network performance, we employed the same approach used for modelling human behaviour (Fig 1E), and investigated suddenness, selectivity, and delay. To identify sudden performance improvements, we fitted each network's time course of accuracy on low coherence trials by (1) a linear model and (2) a non-linear sigmoid function, which would indicate gradual performance increases or insight-like behaviour, respectively. As in humans, network performance on low coherence trials was best fit by a non-linear sigmoid function, indicating at least a subsection of putative "insight networks" (BIC sigmoid function: $M = -10$, $SD = 1.9$, protected exceedance probability: 1, BIC linear function: $M = -9$, $SD = 2.4$, protected exceedance probability: 0)(Fig 3F).

We then tested whether insight-like behaviour occurred only in a subset of networks (selectivity) by assessing in how many networks the steepness of the performance increase exceeded a chance level defined by a baseline distribution of the steepness. As in humans, we ran simulations of 99 control networks with the same architecture, which were trained on the same task except that during the *motion and colour phase*, the two inputs remained uncorrelated. About

half of networks (46/99, 46.5%) had steepness values larger than the 100% percentile of the control distribution, closely matching the value we observed in the human sample. The L1-networks that showed sudden performance improvements were not matched to insight humans more often than chance ($\chi^2(47) = 27.9$, $p = 0.99$), suggesting that network variability did not originate from baseline performance levels or trial orders. Hence, a random subset of networks showed sudden performance improvements comparable to those observed during insight moments in humans (Fig 3G).

For simplicity reasons in comparing network behaviour to humans, we will refer to the two groups as "insight and no-insight networks". Analysing behaviour separately for the insight and no-insight networks showed that switches to the colour strategy improved the networks' performance on the lowest coherence trials once the *motion and colour phase* started, as compared to networks that did not show a strategy shift ($M = 83 \pm 11\%$, vs. $M = 64 \pm 9\%$, respectively, $t(89.8) = 9.2$, $p < .001$, $d = 1.9$, see Fig 3C). The same performance difference between insight and no-insight networks applied when all coherence levels of the *motion and colour phase* were included ($M = 88 \pm 7\%$ vs. $M = 77 \pm 6\%$, $t(93.4) = 7.8$, $p < .001$, $d = 1.57$). Unexpectedly, insight networks performed slightly worse on low coherence trials in the motion phase, i.e. before the change in predictiveness of the features, ($t(97) = -3.1$, $p = 0.003$, $d = 0.62$) (insight networks: $M = 58 \pm 8\%$; no-insight networks: $M = 64 \pm 9\%$), and in contrast to the lack of pre-insight differences we found in humans.

Finally we asked whether insight-like behaviour occurred with random delays in neural networks, again scrutinising the time points of inflection of the fitted sigmoid function, i.e. when performance exhibited abrupt increases (see Methods). Time-locking the data to these individually fitted switch points verified that, as in humans, the insight-like performance increase was particularly evident around the switch points: accuracy was significantly increased between the halved task blocks preceding and following the insight-like behavioural switch, for colour switching networks ($M = 66 \pm 8\%$ vs. $M = 86 \pm 7\%$, $t(91.6) = -12.7$, $p < .001$, $d = 2.6$, see Fig 3C and Fig EB in S1 Text).

Among insight networks, the delay distribution ranged from 2 to 8 trial bins after the start of the *motion and colour phase*, and did not differ from a uniform distribution taking into account the hazard rate (Exact two-sided Kolmogorov-Smirnov test: $D(46) = 0.13$, $p = 0.85$). The average delay of insight-like switches was 3.5 trial bins ($\pm1.05$), corresponding to 175 trials (Fig 3H). The insight networks' delay was thus slightly longer than for humans ($M = 130 \pm 95$ trials vs. $M = 175 \pm 105$ trials, $t(92.7) = -2.1$, $p = 0.04$, $d = 0.42$). The variance of insight-like strategy switch onsets as well as the relative variance in the abruptness of the switch onsets thus qualitatively matched our behavioural results observed in human participants. The behaviour of L1-regularised neural networks therefore showed all characteristics of human insight: sudden improvements in performance that occurred selectively only in a subgroup with variable random delays.

We investigated whether this effect was specific to the form of regularisation and found that neither L2-regularisation nor non-regularised networks showed the insight key behavioural characteristics of selectivity and delay (see Figs H, I and J in S1 Text).

**Spontaneous strategy switches in deep neural networks.** Analysing the results from the models with a more complex network architecture revealed qualitatively similar results. When we applied L1-regularisation with a regularisation parameter of $\lambda = 0.002$ on the gate weights of hidden layer networks, 18.2% of the networks exhibited *abrupt* and *delayed* learning dynamics, resembling insight-like behaviour in humans (Fig A in S1 Text). Insight-like switches to the colour strategy thereby again improved the networks' performance significantly. We also observed a wide distribution of delays, for L1-regularised networks with a hidden layer (Fig AC and AD in S1 Text). Taken together, these results from simple deep neural

networks mirror our observations from simulations with a simplified setup. We can thus confirm that our results of L1-regularised neural networks' behaviour exhibiting all key characteristics of human insight-like spontaneous switches (suddenness, selectivity and delay) are not an artefact of the one-layer linearity.

## Origins of insight-like behaviour in neural networks

Having established the behavioural similarity between L1-networks and humans in an insight task, we asked what gave rise to insight-like switches in some networks, but not others. We therefore investigated the dynamics of gate weights and the effects of noise in insight vs. no-insight networks, and the role of regularisation strength parameter λ.

**Insight networks immediately learn more about colour once it becomes predictive.** Our first question was how learning about stimulus colour differed between insight and no-insight L1-networks, as expressed by the dynamics of network gradients. We time-locked the time courses of gradients to each network's individual switch point. Right when the switch occurred (at t of the estimated switch), colour gate weight gradients were significantly larger in insight compared to no-insight L1-networks ($M = 0.06 \pm 0.06$ vs. $M = 0.02 \pm 0.03$, $t(73.2) = 5.1$, $p < .001$, $d = 1.05$), while this was not true for motion gate weight gradients ($M = 0.18 \pm 0.16$ vs. $M = 0.16 \pm 0.16$, $t(97) = 0.7$, $p = 0.5$, $d = 0.13$).

Notably, insight networks had larger colour gate weight gradients even before any behavioural changes were apparent, right at the beginning of the *motion and colour phase* (first 5 trials of *motion and colour phase*: $M = 0.05 \pm 0.07$ vs. $M = 0.01 \pm 0.01$; $t(320) = 8.7$, $p < .001$), whereas motion gradients did not differ ($t(576.5) = -0.1$, $p = 0.95$). This increase in colour gate weight gradients for insight networks happened within a few trials after correlation onset (colour gradient last trial of *motion phase*: $M = 0 \pm 0$ vs. 5th trial of *motion and colour phase*: $M = 0.06 \pm 0.08$; $t(47) = -5.6$, $p < .001$, $d = 1.13$), and suggests that insight networks start early to silently learn more about colour inputs compared to their no-insight counterparts. A change point analysis considering the mean and variance of the gradients confirmed the onset of the *motion and colour phase* to be the change point of the colour gradient mean, with a difference of 0.04 between the consecutive pre-change and change time points for insight networks vs 0.005 for no-insight networks (with a change point detected two trials later), indicating considerable learning about colour for insight networks.

**"Silent" colour knowledge precedes insight-like behaviour.** A core feature of our network architecture is that inputs were multiplied by two factors, a gate *g*, and a weight *w*, but only gates were regularised. This meant that some networks might have developed larger colour weights, but still showed no signs of colour use, because the gates were very small. This could explain the early differences in gradients reported above. To test this idea, we investigated the absolute size of colour gates and weights of insight vs no-insight L1-networks before and after insight-like switches had occurred.

Comparing gates at the start of learning (first trial of the *motion and colour phase*), there were no differences between insight and no-insight networks for either motion or colour gates (colour gates: $M = 0 \pm 0.01$ vs. $M = 0 \pm 0.01$; $t(95.3) = 0.8$, $p = 0.44$, motion gates: $M = 0.5 \pm 0.3$ vs. $M = 0.6 \pm 0.3$; $t(93.1) = -1.7$, $p = 0.09$, see Fig 4A, 4F and 4H). Around the individually fitted switch points, however, the gates of insight and no-insight networks differed only for colour gates (colour gates: $0.2 \pm 0.2$ vs $0.01 \pm 0.02$ for insight vs no-insight networks, $t(48) = 6.7$, $p < 0.001$, $d = 1.4$, motion gates: $0.5 \pm 0.3$ vs $0.5 \pm 0.3$ for insight vs no-insight networks, $t(95.6) = 0.2$, $p = 0.9$, $d = 0.04$. Insight networks' increased use of colour inputs was particularly evident at the end of learning (last trial of the *motion and colour phase*) and reflected in larger colour gates ($0.7 \pm 0.3$ vs $0.07 \pm 0.2$ for insight vs no-insight networks, $t(73.7) = 13.4$,

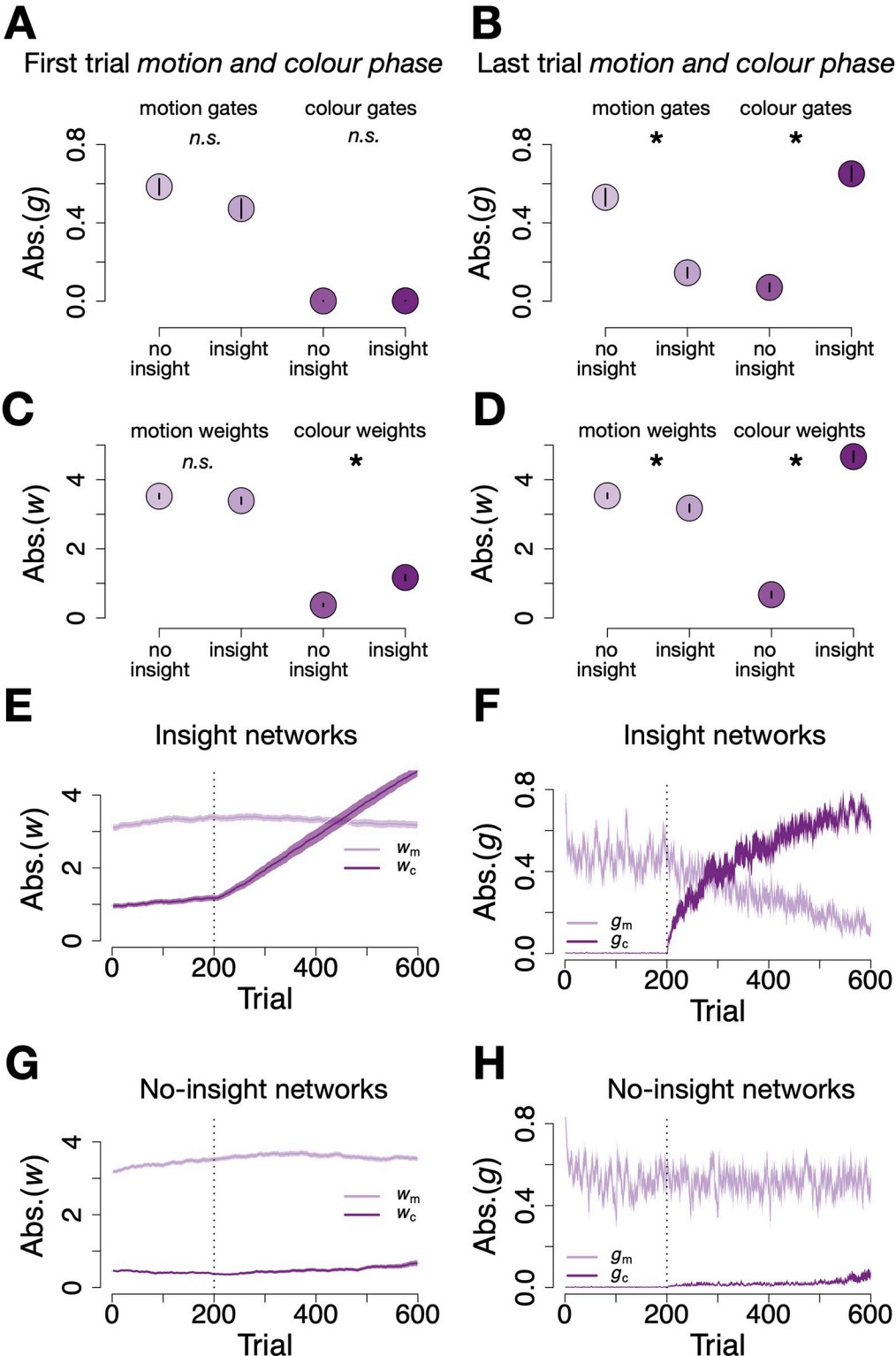

**Fig 4. Differences between insight and no-insight networks in parameter dynamics.** Absolute average magnitude of motion (light purple) and colour (dark purple) gates at the first trial **(A)** and the last trial **(B)** of the *motion and colour phase*, shown separately for insight and no-insight networks. Differences between the gates of insight vs no-insight networks emerged only at the end of *motion and colour phase*, i.e. only after insight-like behaviour had occurred. Panels **(C)** and **(D)** show the same for absolute weight magnitudes. Unlike gates, colour weights differed between network types

already before insight-like switches were detectable, i.e. at the start of the *motion and colour phase*. Both absolute colour weight (**E**) and gate (**F**) magnitudes increase after start of the *motion and colour phase* (trial 200, dashed vertical line) for insight networks, while absolute colour gates decrease in magnitude after colour correlation onset. For no-insight networks absolute weight (**G**) and gate (**H**) magnitudes do not show such dynamics and remain stable after the onset of the *motion and colour phase*. Error bars/shadows signify SEM.

$p < 0.001$, $d = 2.7$) while the reverse was true for motion gates ($M = 0.2 \pm 0.2$ vs $M = 0.5 \pm 0.3$, respectively, $t(81) = -7.5$, $p < 0.001$, $d = 1.5$, see Fig 4B, 4F and 4H). Hence, differences in gating between network subgroups were only present after, but not before learning, and did not explain the above reported gradient differences or which network would show insight-like behaviour.

A different pattern emerged when investigating the weights of the networks. Among insight networks colour weights were significantly larger already at the start of learning (first trial of the *motion and colour phase*), as compared to no-insight networks (insight: $M = 1.2 \pm 0.6$; no-insight: $M = 0.4 \pm 0.3$, $t(66.2) = 8.1$, $p < .001$, $d = 1.7$, see Fig 4C, 4E and 4G). This was not true for motion weights (insight: $M = 3.4 \pm 0.7$; no-insight: $M = 3.5 \pm 0.5$, $t(89.5) = -1.1$, $p = 0.3$, $d = 0.2$, see Fig 4C, 4E and 4G). Thus, colour information appeared to be encoded in the weights of insight networks already before any insight-like switches occurred. Because the colour gates were suppressed through the L1-regularisation mechanism before learning, the networks did not differ in any observable colour sensitivity. An increase of colour gates reported above could then unlock the "silent knowlegde" of colour relevance.

To experimentally test the effect of pre-learning colour weights, we ran a new sample of L1-networks ($N = 99$), and adjusted the colour and motion weight of each respective network to the mean absolute colour and motion weight size we observed in insight networks at the start of learning (first trial of *motion and colour phase*). Gates were left untouched. This increased the number of insight networks from 46.5% to 70.7%, confirming that encoding of colour information at an early stage was an important factor for later switches, but also not sufficient to cause insight-like behaviour in all networks. Note that before weights adjustments were made, the performance of the new networks did not differ from the original L1-networks ($M = 0.8 \pm 0.07$ vs $M = 0.8 \pm 0.07$, $t(195) = 0.2$, $p = 0.9$, $d = 0.03$). In our new sample, networks that would later show insight-like behaviour or not also did not differ from each other (insight: $M = 0.7 \pm 0.07$ vs $M = 0.7 \pm 0.07$, $t(100.9) = 1.4$, $p = 0.2$, $d = 0.3$, no-insight: $M = 0.8 \pm 0.05$ vs $M = 0.8 \pm 0.07$, $t(71) = 0.9$, $p = 0.4$, $d = 0.2$). Weight and gate differences between L1- and L2-networks are reported in the Supplementary Material (Fig I in S1 Text).

**Networks need noise for insight-like behaviour.**   One possible factor that could explain the early differences between the weights of network subgroups is noise. The networks were exposed to noise at two levels: on each trial noise was added at the output stage ($\eta \sim \mathcal{N}(0, \sigma_\eta^2)$), and to the gate and weight gradients during updating ($\xi \sim \mathcal{N}(0, \sigma_\xi^2)$).

We probed whether varying the level of noise added during gradient updating, i.e. $\sigma_\xi$, would affect the proportion of networks exhibiting insight-like behaviour. Parametrically varying the variance of noise added to colour and motion gates and weights led to increases in insight-like behaviour, from no single insight network when no noise was added to 100% insight networks when $\sigma_{\xi_g}$ reached values of larger than approx. 0.05 (Fig 5A). Since gate and weight updates were coupled (see Eqs 4–7), noise during one gradient update could in principle affect other updates as well. We therefore separately manipulated the noise added to updates of colour gates and weights, motion gates and weights, all weights and all gates. This showed that adding noise to only weights during the updates was sufficient to induce insight-like behaviour (Fig 5B). In principle, adding noise to only gates was sufficient for insight-like

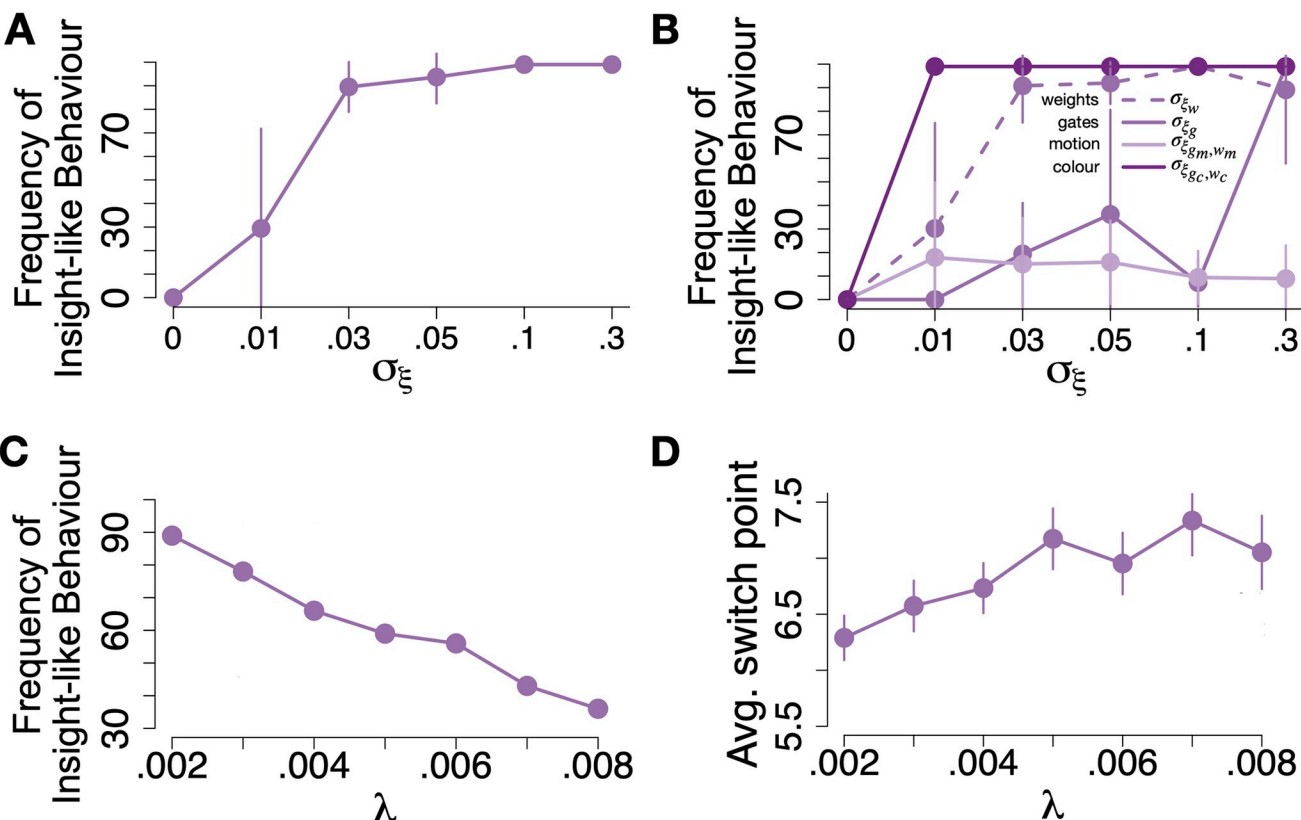

**Fig 5.** Influence of gradient noise $\sigma_\xi$ and regularisation $\lambda$ on insight-like switches **(A)** Influence of gradient noise (standard deviation $\sigma_\xi$) on the frequency of absolute numbers of insight-like networks. The frequency of insight-like switches increases gradually with $\sigma_\xi$ until it reaches a ceiling around $\sigma_\xi = .05$. Error bars are SD. Results from 10 simulations of 99 networks each. **(B)** Effects of gradient noise added only to either all weights ($\sigma_{\xi_w}$, light purple dashed line), all gates ($\sigma_{\xi_g}$, solid purple line), all motion parameters (i.e. motion weight and motion gates, $\sigma_{\xi_{g_m,w_m}}$, light purple solid line) and all colour parameters ($\sigma_{\xi_{g_c,w_c}}$, dark purple solid line) on the frequency of insight-like switches. While only small amounts of noise in the colour parameters $\sigma_{\xi_{g_c,w_c}}$ suffice to induce 100% insight-like strategy switches among networks (dark purple line), adding noise to the motion parameters $\sigma_{\xi_{g_c,w_c}}$ (light purple solid line) only had a a very minor effect. Furthermore, we find that adding noise specifically to the weights ($\sigma_{\xi_w}$, dashed purple line), has a much larger effect than adding noise to the gates ($\sigma_{\xi_g}$, solid purple line). Error bars are SD. Results from 10 simulations of 99 networks each. Colour scheme as in Fig 1B **(C)** The frequency of insight-like switches declines with increasing $\lambda$. **(D)** The average switch point occurs later in the task with increasing $\lambda$. Error bars signify SEM.

switches as well, although noise applied to the gates had to be relatively larger to achieve the same effect as applying noise to weight gradients (Fig 5B), presumably due the effect of regularisation. Adding noise only to the gradients of motion gates or weights was not sufficient to induce insight-like switches (Fig 5B). On the other hand, noise added only to the colour parameter updates quickly led to substantial amounts of insight-like behavioural switches (Fig 5B).

An analysis of *cumulative* noise showed that the effects reported above are mostly about momentary noise fluctuations: cumulative noise added to the output did not differ between insight and no-insight networks at either the start (first trial of the *motion and colour phase*) or end of learning (last trial of the *motion and colour phase*) (start: $M = -0.3 \pm 4.7$ vs. $M = -0.6 \pm 3.9$; $t(91.2) = 0.4$, $p = 0.7$, end: $M = 0.6 \pm 7.1$ vs. $M = 0.5 \pm 7.1$; $t(96.7) = 0.07$, $p = 1$), and the same was true for cumulative noise added during the gradient updates to weights and gates (see S1 Text for details).

We therefore conclude that Gaussian noise added to updates of particularly colour gate weights, in combination with "silent knowledge" about colour information stored in suppressed weights, is a crucial factor for insight-like behavioural changes.

**Regularisation parameter λ affects insight behaviour delay and frequency.**   In our previous results, the regularisation parameter λ was arbitrarily set to 0.07. We next tested the effect of of λ on insight-like behaviour. The number of L1-regularised insight networks linearly decreased with increasing λ (Fig 5C). Lambda further had an effect on the delay of the insight-like switches, with smaller λ values leading to decreased average delays of switching to a colour strategy after predictiveness of the inputs had changed (Fig 5D). The regularisation parameter λ thus affects two of the key characteristics of human insight—selectivity and delay.

## Discussion

We investigated insight-like learning behaviour in humans and neural networks. In a binary decision making-task with a hidden regularity that entailed an alternative way to solve the task more efficiently, a subset of L1-regularised neural networks with multiplicative gates of their input channels as an attention mechanism displayed spontaneous, jump-like learning that signified the sudden discovery of the hidden regularity—insight moments often deemed "mysterious" boiled down to the simplest expression. Networks exhibited all key characteristics of human insight-like behaviour in the same task (suddenness, selectivity, delay). Crucially, neural networks were trained with standard online stochastic gradient descent that is often associated with gradual learning. Our results therefore suggest that the behavioural characteristics of aha-moments can arise from gradual learning mechanisms, and hence suffice to mimic human insight. To our knowledge, this is the first time that insight-like behaviour has been shown in a gradual delta rule learning system, devoid of complex cognitive processes.

Network analyses identified the factors which caused insight-like behaviour in L1-networks: noise added during the gradient computations accumulated to non-zero weights in some networks. As long as colour information was not useful yet, i.e. prior to the onset of the hidden regularity, close-to-0 colour gates rendered these weights "silent", such that no effects on behaviour can be observed. Once the hidden colour regularity became available, the non-zero colour weights helped to trigger non-linear learning dynamics that arise during gradient updating, and depend on the starting point. Hence, our results hint at important roles of gating as an "attentional" mechanism, noise, and L1-regularisation as the computational origins of sudden, insight-like behavioural changes. We report several findings that are in line with this interpretation: addition of gradient noise $\xi$ in particular to the colour weights and gates, pre-learning adjustment of colour weights and a reduction of the regularisation parameter λ all increased insight-like behaviour. We note that our networks did not have a hidden layer, witnessing the fact that no hidden layer is needed to produce non-linear learning dynamics. This is however not a necessity, as we can reproduce the same results in more complex networks with a hidden layer (see Fig A in S1 Text).

Our choice to include explicit regularisation of only the gates was motivated by the idea that capacity constraints enforce regularisation only during the output stage, while the initial sensory processing is unaffected. Note, however, that regularising only the weights, but not the gates, would have functionally equivalent effects, and regularisation of both suppresses any learning.

Furthermore, since our architecture is functionally equivalent to a 2-layer diagonal linear network with L1-regularisation on the second layer, our results could equally pertain to a particular simple form of a linear network with 2 layers.

Our findings have implications for the conception of insight phenomena in humans. While present-day machines clearly do not have the capacity to have aha-moments due to their lack of meta-cognitive awareness, our results show that the remarkable behavioural signatures of insight-like strategy switches by themselves do not necessitate a dedicated process. This raises the possibility that sudden behavioural changes which occur even during gradual learning could in turn lead to the subjective effects that accompany insights [58, 59].

It is worth emphasising that, like the commonly used Number Reduction Task (NRT) [54], our task does not mention the possibility of a hidden strategy in the instructions, thus differing from cognitive control paradigms involving deliberate attention switching tasks. One difference in task structure between our task and the NRT is the onset of the hidden rule. In the NRT, the alternative strategy is present from the start, while in our task, the *motion phase* persists for the first 45% of the trials. The rationale behind this is that we do not use explicit instructions in our paradigm, requiring participants to learn the feature relevance through trial-and-error learning. If colour was predictive from the start, subjects would start using it right away, since it is significantly easier than the noisy motion feature. In the NRT, participants are explicitly instructed with a certain task rule. Both tasks thus involve a phase of learning a certain rule at first that can be overcome by insight about different contingencies in the environment.

We note that, besides the Number Reduction Task, our task differs from other common insight paradigms in one important aspect. Usually, subjects are actively trying to find a solution to a certain problem which then suddenly rises to consciousness after experiencing a period of being stuck (impasse) [21]. Here, subjects learn a certain strategy and are unaware that task contingencies will change during the task, which then offers a hidden, more efficient strategy to solve the task that can be discovered through insight. However, the behavioural signature of insight in our task paradigm shares the fundamental characteristic of happening suddenly and after a variable delay. We regard the latter as analogous to impasse since subjects are blind to the alternative solution during that period. Based on these behavioural findings, we conclude that albeit differing from classic insight paradigms [8, 9, 57], our task captures a special case of the general insight phenomenon. Further, insight has recently been conceived as a core cognitive mechanism of a unifying framework accounting for changes in mental representations, that reach beyond problem solving and include sub-fields as various as psychotherapy, psychedelic research and meditation [23]. Our neural network model also differs from other computational accounts of insight in the same way that our task neither includes a classic insight problem to actively solve, nor an off-task incubation period. The EII (explicit-implicit interaction) model [28] for example uses the CLARION architecture which has two separate, but interacting modules for explicit and implicit knowledge. The insight goal in this account is reached when an internal confidence level threshold is crossed through an active, iterative problem solving process. Our model architecture differs from this as information is represented and processed in one single way. Insight-like behaviour is furthermore a phenomenon occurring "for free" at the same time that gradual learning happens in our task which stands in contrast to the EII model which defines insight, not performance, as the ultimate goal.

Our results highlight noise and regularisation as aspects of brain function that are involved in the generation of insights. Cellular and synaptic noise is omnipresent in brain activity [60, 61], and has a number of known benefits, such as stochastic resonance and robustness that comes with probabilistic firing of neurons based on statistical fluctuations due to Poissonian neural spike timing [62]. It has also been noted that noise plays an important role in jumps between brain states, when noise provokes transitioning between attractor states [63]. Previous studies have therefore noted that stochastic brain dynamics can be advantageous, allowing e.g. for creative problem solving (as in our case), exploratory behaviour, and accurate decision

making [60, 61, 63, 64]. Albeit our conceptual model was not meant to be biologically realistic, this work adds a computationally precise explanation of how noise can lead to insight-like behaviour to this literature. Questions about whether inter-individual differences in neural variability predict insights [64], or about whether noise that occurs during synaptic updating is crucial remain an interesting topic for future research.

While our simulations focused on the specific explicit regularisation mechanism of adding a penalty term to the error function, many other forms of implicit regularisation, such as weight sharing, might exist and be possibly implemented in the brain. To what extent our results generalise to such other regularisation techniques is unknown and speculative at this point. Regularisation has for instance been implied in synaptic scaling, which helps to adjust synaptic weights in order to maintain a global firing homeostasis [65], thereby aiding energy requirements and reducing memory interference [66, 67]. It has also been proposed that regularisation modulates the threshold for induction of long-term potentiation [65]. These mechanisms therefore present possible synaptic factors that contribute to insight-like behaviour in humans and animals. We note that synaptic scaling has often been linked to sleep [66], and regularisation during sleep has also been suggested to help avoid overfitting to experiences made during the day, and therefore generalisation [68].

A follow up study to this experiment that investigated the effect of a daytime nap intervention on insight [69], found that the steepness of the spectral slope during sleep, which has been indirectly linked to regularisation [70], predicted insight above and beyond sleep stages. The relationship between insight-like behaviour and regularisation in the broader sense—and different regularisation mechanisms specifically—thus remains an interesting area for future research.

On a more cognitive level, regularisation has been implied in the context of heuristics. In this notion, regularisation has been proposed to function as an infinitely strong prior in a Bayesian inference framework [42]. This infinitely strong prior would work as a sort of attention mechanism and regularise input and information in a way that is congruent with the specific prior, whereas a finite prior would under this assumption enable learning from experience [42]. Another account regards cognitive control as regularised optimisation [71]. According to this theory, better transfer learning is supported by effort costs regularising towards more task-general policies. It therefore seems possible that the factors that impact regularisation during learning can also lead to a neural switch between states that might be more or less likely to govern insights.

The occurrence of insight-like behaviour with the same characteristics as found in humans was specific to L1-regularised networks, while no comparable similarity occurred in L2- or non-regularised networks. While no hidden layer is necessary to produce this result, the same L1-specific effect can be replicated in a model with a hidden layer (Fig A in S1 Text). Although L2-regularised neural networks learned to suppress initially irrelevant colour feature inputs and showed abrupt performance increases reminiscent of insights, only L1 networks exhibited a wide distribution of time points when the insight-like switches occur (delay) as well as a selectivity of the phenomenon to a subgroup of networks, as found in humans. We note that L2- and non-regularised networks technically performed better on the task, because they collectively improve their behavioural efficiency sooner. One important question therefore remains under which circumstances L1 would be the most beneficial form of regularisation. One possibility could be that the task is too simple for L1-regularisation to be beneficial. It is conceivable that L1-regularisation only starts being advantageous in more complex task settings when generalisation across task sets is required and a segregation of task dimensions to learn about at a given time would prove useful.

Taken together, gradual training of neural networks with gate modulation leads to insight-like behaviour as observed in humans, and points to roles of regularisation, noise and "silent knowledge" in this process.

While general scalability remains an issue for future research, we believe that the mechanistic insights from our model have implications for more complex systems, such as the brain. Our results imply a link between behavioural markers of insight and noise measurements as well as regularisation in the brain. Towards this goal of generalisation we replicated our results using a multi-neuron, deep, non-linear model with a 48-unit hidden layer and investigated effects of different types of explicit regularisation (L1 vs L2 vs no regularisation), showing that only L1-regularised networks exhibit all three key characteristics of human insight. We further explained that our network is identical to a 2-layer diagonal network with L1-regularisation on the second layer and that L1-regularisation of weights has equivalent effects of gate regularisation, while L1-regularising both entirely changes the model behaviour. We speculate that even more complex, deeper networks, which would allow to disentangle several factors of variation (e.g. motion, shape, colour, lighting) and apply gating variables to these, might exhibit similar shifts between relying on subsets of these features.

These results make an important contribution to the general understanding of learning dynamics and representation formation in environments with non-stationary feature relevance in both biological and artificial agents.

## Methods

### Ethics statement

The study protocol was approved by the local ethics committee of the Max Planck Institute for Human Development (approval number N-2020-03). All participants gave informed consent prior to beginning the experiment.

### Task

**Stimuli.**   We employed a perceptual decision task that required a binary choice about circular arrays of moving dots [56], similar to the Spontaneous Strategy Switch Task developed earlier [13]. Dots were characterised by two features, (1) a motion direction (four possible orthogonal directions: NW, NE, SW, SE) and (2) a colour (orange or purple, Fig 1A). The noise level of the motion feature was varied in 5 steps (5%, 10%, 20%, 30% or 45% coherent motion), making motion judgement relatively harder or easier. Colour difficulty was constant, thus consistently allowing easy identification of the stimulus colour. The condition with most noise (5% coherence) occurred slightly more frequently than the other conditions (30 trial per 100, vs 10, 20, 20, 20 for the other conditions).

The task was coded in JavaScript and made use of the jsPsych 6.1.0 plugins. Participants were restricted to use desktops (no tablets or mobile phones) of at least 13 inch width diagonally. Subjects were further restricted to use either a Firefox or Google Chrome browser to run the experiment.

On every trial, participants were presented a cloud of 200 moving dots with a radius of 7 pixels each. In order to avoid tracking of individual dots, dots had a lifetime of 10 frames before they were replaced. Within the circle shape of 400 pixel width, a single dot moved 6 pixel lengths in a given frame. Each dot was either designated to be coherent or incoherent and remained so throughout all frames in the display, whereby each incoherent dot followed a randomly designated alternative direction of motion.

The trial duration was 2000 ms and a response could be made at any point during that time window. After a response had been made via one of the two button presses, the white fixation

cross at the centre of the stimulus would turn into a binary feedback symbol (happy or sad smiley) that would be displayed until the end of the trial (Fig 1C). An inter trial interval (ITI) of either 400, 600, 800 or 1000 ms was randomly selected. If no response was made, a "TOO SLOW" feedback was displayed for 300 ms before being replaced by the fixation cross for the remaining time of the ITI.

**Task design.** For the first 400 trials, the *motion phase*, the correct binary choice was only related to stimulus motion (two directions each on a diagonal were mapped onto one choice), while the colour changed randomly from trial to trial (Fig 1B). For the binary choice, participants were given two response keys, "X" and "M". The NW and SE motion directions corresponded to a left key press ("X"), while NE and SW corresponded to a right key press ("M") (Fig 1A). Participants received trial-wise binary feedback (correct or incorrect), and therefore could learn which choice they had to make in response to which motion direction (Fig 1C).

We did not specifically instruct participants to pay attention to the motion direction. Instead, we instructed them to learn how to classify the moving dot clouds using the two response keys, so that they would maximise their number of correct choices. To ensure that participants would pick up on the motion relevance and the correct stimulus-response mapping, motion coherence was set to be at 100% in the first block (100 trials), meaning that all dots moved towards one coherent direction. Participants learned this mapping well and performed close to ceiling (87% correct, t-test against chance: $t(98) = 37.4$, $p < .001$). In the second task block, we introduced the lowest, and therefore easiest, three levels of motion noise (20%, 30% and 45% coherent motion), before starting to use all five noise levels in block 3. Since choices during this phase should become solely dependent on motion, they should be affected by the level of motion noise. We assessed how well participants had learned to discriminate the motion direction after the fourth block. Participants that did not reach an accuracy level of at least 85% in the three lowest motion noise levels during this last task block of the pretraining were excluded from the *motion and colour phase*. The 85% accuracy threshold on low motion noise trials was based on previous studies, where subjects' performance was at a comparable level [13, 14, 16]. Before starting the experiment all subjects were notified, that they could only advance to the second task phase (*motion and colour phase*, although this was not communicated to participants) if they performed well enough in the first phase and that they would be paid accordingly for either one or two completed task phases. Based on the early stopping of the experiment for participants who performed below threshold in the *motion phase*, a post-hoc change of the performance criterion is not possible. Results from other studies using this or a similar task imply that the reported insight phenomenon is not an artefact of the early stopping threshold. [13–15, 69]. After the *motion phase*, in the *motion and colour phase*, the colour feature became predictive of the correct choice in addition to the motion feature (Fig 1B). This meant that each response key, and thus motion direction diagonal, was consistently paired with one colour, and that colour was fully predictive of the required choice. Orange henceforth corresponded to a correct "X" key press and a NW/SE motion direction, while purple was predictive of a correct "M" key press and NE/SW motion direction (Fig 1A). This change in feature relevance was not announced to participants, and the task continued for another 400 trials as before—the only change being the predictiveness of colour.

Before the last task block we asked participants whether they 1) noticed a rule in the experiment, 2) how long it took until they noticed it, 3) whether they used the colour feature to make their choices and 4) to replicate the mapping between stimulus colour and motion directions. We then instructed them about the correct colour mapping and asked them to rely on colour for the last task block. This served as a proof that subjects were in principle able to do the task based on the colour feature and to show that, based on this easier task strategy, accuracy should be near ceiling for all participants in the last instructed block.

## Human participants

Participants between eighteen and 30 years of age were recruited online through Prolific.

Participation in the study was contingent on showing learning of the stimulus classification. Hence, to assess whether participants had learned to correctly identify motion directions of the moving dots, we probed their accuracy on the three easiest, least noisiest coherence levels in the last block of the uncorrelated task phase. If subjects reached an accuracy level of at least 85%, they were selected for participation in the experiment.

Ninety-six participants were excluded due to insufficient accuracy levels after the *motion phase* as described above. 99 participants learned to classify the dots' motion direction, passed the accuracy criterion and completed both task phases. These subjects make up the final sample included in all analyses. Note that in previous studies where we did not apply such a criterion, similar spontaneous strategy switch behaviour was observed [13, 17, 20]. 34 participants were excluded due to various technical problems or premature quitting of the experiment. Participants received 3£ for completing only the first task phase and 7£ for completing both task phases.

## Neural networks

**L1-regularised neural networks.** We utilise a simple neural network model to reproduce the observations of the human behavioural data in a simplified supervised learning regression setting. We trained a simple neural network with two input nodes, two input gates and one output node on the same decision making task (Fig 1D).

The network received two inputs, $x_m$ and $x_c$, corresponding to the stimulus motion and colour, respectively, and had one output, $\hat{y}$. Importantly, each input had one associated multiplicative gate ($g_m, g_c$) such that output activation was defined as $\hat{y} = \text{sign}(g_m w_m x_m + g_c w_c x_c + \eta)$ where $\eta \sim \mathcal{N}(0, \sigma)$ is Gaussian noise (Fig 1D).

To introduce competitive dynamics between the input channels, we added L1-regularisation on the gate weights $g$, resulting in the following loss function:

$$\mathcal{L} = \frac{1}{2}\left(g_m w_m x_m + g_c w_c x_c + \eta - y\right)^2 + \lambda(|g_m| + |g_c|) \tag{5}$$

The network was trained in a gradual fashion through online gradient descent with Gaussian white noise $\xi$ added to the gradient update and a fixed learning rate $\alpha$. Given the loss function, this yields the following update equations for noisy stochastic gradient descent (SGD):

$$\Delta w_m = -\alpha x_m g_m (x_m g_m w_m + x_c g_c w_c + \eta - y) + \xi_{w_m} \tag{6}$$

$$\Delta g_m = -\alpha x_m w_m (x_m g_m w_m + x_c g_c w_c + \eta - y) - \alpha\lambda\,\text{sign}(g_m) + \xi_{g_m} \tag{7}$$

$$\Delta w_c = -\alpha x_c g_c (x_c g_c w_c + x_m g_m w_m + \eta - y) + \xi_{w_c} \tag{8}$$

$$\Delta g_c = -\alpha x_c w_c (x_c g_c w_c + x_m g_m w_m + \eta - y) - \alpha\lambda\,\text{sign}(g_c) + \xi_{g_c} \tag{9}$$

with $\lambda = 0.07$, $\alpha = 0.6$ and $\sigma_\xi = 0.05$).

This implies that the evolution of the colour weights and gates will exhibit non-linear quadratic and cubic dynamics, driven by the interaction of $w_c$ and $g_c$. Multiplying the weights $w$ with the regularised gate weights $g$ leads to smaller weights and therefore initially slower increases of the colour weights $w_c$ and respective gate weights $g_c$ after colour has become predictive of correct choices.

To understand this effect of non-linearity analytically, we used a simplified setup of the same model without gate weights:

$$\mathcal{L} = \left[ w_m x_m + w_c x_c + \eta - y \right]^2 \tag{10}$$

Using this model, we observe exponential increases of the colour weights $w_c$ after the onset of the *motion and colour phase*. This confirms that the interaction of $w_c$ and $g_c$, as well as the regularisation applied to $g_c$ are necessary for the insight-like non-linear dynamics including a distribution of insight-like strategy switch onsets as well as variety in slope steepness of insight-like switches.

The accuracy is given by:

$$\mathbb{P}[\hat{y} = y | w_m, g_m, w_c, g_c]$$
$$= \frac{1}{2} \left[ 1 + \text{erf} \left( \frac{g_m w_m x_m + g_c w_c x_c}{\sqrt{2 \left( (g_m w_m \sigma_m)^2 + (g_c w_c \sigma_c)^2 + \sigma^2 \right)}} \right) \right] \tag{11}$$

We trained the network on a curriculum precisely matched to the human task, and adjusted hyperparameters (noise levels), such that baseline network performance and learning speed were carefully equated between humans and networks.

Specifically, we simulated the same number of networks than humans were included in the final analysis sample ($N = 99$). We matched the motion noise based performance variance of a given simulation to a respective human subject using a non-linear COBYLA optimiser. While the mean of the colour input distribution (0.22) as well as the standard deviations of both input distributions were fixed (0.01 for colour and 0.1 for motion), the respective motion input distribution mean values were individually fitted for each single simulation as described above.

The input sequences the networks received were sampled from the same ten input sequences that humans were exposed to in task phase two. This means that for the task part where colour was predictive of the correct binary choice, *motion and colour phase* (500 trials in total), networks and humans received the same input sequences.

The networks were given a slightly longer *training phase* of six blocks (600 trials) in comparison to the two blocks *training phase* that human subjects were exposed to (Fig 1B). Furthermore, human participants first completed a block with 100% motion coherence before doing one block with low motion noise. The networks received six *training phase* blocks containing the three highest motion coherence levels. Both human subjects and networks completed two blocks including all noise levels in the *motion phase* before colour became predictive in the *motion and colour phase*.

**L2-regularised neural networks.**   To probe the effect of the aggressiveness of the regulariser on insight-like switch behaviour in networks, we compared our L1-regularised networks with models of the same architecture, but added L2-regularisation on the gate weights *g*. This yielded the following loss function:

$$\mathcal{L} = \frac{1}{2} \left( g_m w_m x_m + g_c w_c x_c + \eta - y \right)^2 + \frac{\lambda}{2} \left( |g_m| + |g_c| \right)^2 \tag{12}$$

From the loss function we can again derive the following update equations for noisy stochastic gradient descent (SGD):

$$\Delta w_m = -\alpha x_m g_m (x_m g_m w_m + x_c g_c w_c + \eta - y) + \xi_{w_m} \tag{13}$$

$$\Delta g_m = -\alpha x_m w_m (x_m g_m w_m + x_c g_c w_c + \eta - y) - \alpha \lambda g_m + \xi_{g_m} \tag{14}$$

$$\Delta w_c = -\alpha x_c g_c (x_c g_c w_c + x_m g_m w_m + \eta - y) + \xi_{w_c} \tag{15}$$

$$\Delta g_c = -\alpha x_c w_c (x_c g_c w_c + x_m g_m w_m + \eta - y) - \alpha \lambda g_c + \xi_{g_c} \tag{16}$$

with $\lambda = 0.07$, $\alpha = 0.6$ and $\sigma_\xi = 0.05$).

The training is otherwise the same as for the L1-regularised networks.

## Modelling of insight-like switches

**Models of colour use.**   In order to probe whether strategy switches in low coherence trials occurred abruptly, we compared three different models with different assumptions about the form of the data. First, we fitted a linear model with two free parameters:

$$y = m_t + y_0 \tag{17}$$

where $m$ is the slope, $y_0$ the y-intercept and $t$ is time (here, task blocks)(Fig C in S1 Text). This model should fit no-insight participants' data well where colour use either increases linearly over the course of the experiment or stays at a constant level.

Contrasting the assumptions of the linear model, we next tested whether colour-based responses increased abruptly by fitting a step model with three free parameters, a switch point $t_s$, the step size $s$ and a maximum value $y_{max}$ (Fig B in S1 Text), so that

$$y = \begin{cases} y_{max} - s & \text{if } t < t_s \\ y_{max} & \text{if } t \geq t_s \end{cases} \tag{18}$$

We also included a sigmoid function with three free parameters as a smoother approximation of the step model:

$$y = \frac{y_{max} - y_{min}}{1 + e^{-m(t-t_s)}} + y_{min} \tag{19}$$

where $y_{max}$ is the fitted maximum value of the function, $m$ is the slope and $t_s$ is the inflection point (Fig B in S1 Text). $y_{min}$ was given by each individual's averaged accuracy on 5% motion coherence trials in block 3–4.

Comparing the model fits across all subjects using the Bayesian Information Criterion (BIC) and protected exceedance probabilities yielded a preference for the sigmoid function over both a step and linear model, for both humans (Fig 2E) and L1-regularised neural networks (Fig 3D). On the one hand, this supports our hypothesis that insight-like strategy switches do not occur in an incremental linear fashion, but abruptly, with variance in the steepness of the switch. Secondly, this implies that at least a subset of subjects shows evidence for an insight-like strategy switch.

**Human participants.**   To investigate these insight-like strategy adaptations, we modelled human participants' data using the individually fitted sigmoid functions (Fig C in S1 Text). The criterion we defined in order to assess whether a subject had switched to the colour strategy, was the slope at the inflection point, expressing how steep the performance jump was after

having an insight about colour (Fig 1E). We obtained this value by taking the sigmoid function's partial derivative of time

$$\frac{\partial y}{\partial t} = (y_{max} - y_{min})\frac{me^{-m(t-t_s)}}{\left(1 + e^{-m(t-t_s)}\right)^2} \tag{20}$$

and then evaluating the above equation for the fitted switch point, $t = t_s$, which yields:

$$y'(t_s) = \frac{1}{4}m(y_{max} - y_{min}) \tag{21}$$

Switch misclassifications can happen that are caused by irregularities and small jumps in the data—irrespective of a colour strategy switch. We therefore corrected for a general fit of the data to the model by subtracting the individually assessed general model fit from the slope steepness at the inflection point. Insight subjects were then classified as those participants whose corrected slope steepness at inflection point parameters were outside of the 100% percentile of a control group's (no change in predictiveness of colour) distribution of that same parameter (Fig 1E). By definition, insights about a colour rule cannot occur in this control condition, hence our derived out-of-sample distribution evidences abrupt strategy improvements hinting at insight (Fig 3F).

Before the last task block we asked participants whether they used the colour feature to make their choices. 57.6% of participants indicated that they used colour to press correctly. The 49.5% insight participants we identified using our classification method overlapped to 79.6% with participants' self reports (Fig 1E).

**Neural networks.**   We used the same classification procedure for neural networks. All individual sigmoid function fits for L1-regularised networks can be found in the S1 Text (Fig D in S1 Text).

## Supporting information

**S1 Text. Supporting text.** Supplementary information file including additional analyses on behavioural effects without feedback and late onset of low coherence trials, effects of regularisation type on network behaviour, the hidden layer model, weight and gate differences between L1- and L2-regularised networks as well as Gaussian noise differences at weights and gates between insight and no-insight networks. This file includes Fig A (Switch-aligned performance and switch point distributions for L1- and L2-regularised neural networks with a 48 unit hidden layer each), Fig B (Illustrations of models and respective parameters), Fig C (Performance on highest motion noise trials and model predictions for every human participant), Fig D (Performance on highest motion noise trials and model predictions for every network), Fig E (Switch-aligned performance for insight group and no-insight group), Fig F (Switch-aligned performance and overlap between classification and self-reported colour use), Fig G (Trial-wise insight-like strategy improvements for 5% motion coherence trials), Fig H (L2 networks: Task performance and insight-like strategy switches) and Fig J (Comparison of gate weight magnitude and influence of λ on insight-like behaviour across L1- and L2-regularised networks). Figure legends see inside S1 Text.
(PDF)

## Acknowledgments

We thank Robert Gaschler for helpful comments on this manuscript.

## Author Contributions

**Conceptualization:** Anika T. Löwe, Paul S. Muhle-Karbe, Andrew M. Saxe, Christopher Summerfield, Nicolas W. Schuck.

**Formal analysis:** Anika T. Löwe, Léo Touzo, Andrew M. Saxe, Nicolas W. Schuck.

**Funding acquisition:** Nicolas W. Schuck.

**Investigation:** Anika T. Löwe.

**Methodology:** Anika T. Löwe, Nicolas W. Schuck.

**Project administration:** Nicolas W. Schuck.

**Supervision:** Andrew M. Saxe, Christopher Summerfield, Nicolas W. Schuck.

**Visualization:** Anika T. Löwe, Nicolas W. Schuck.

**Writing – original draft:** Anika T. Löwe, Christopher Summerfield, Nicolas W. Schuck.

**Writing – review & editing:** Anika T. Löwe, Léo Touzo, Paul S. Muhle-Karbe, Andrew M. Saxe, Christopher Summerfield, Nicolas W. Schuck.

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
