## [Decision Letter · Decision Letter 0]

14 Jun 2024

Dear Schuck,

Thank you very much for submitting your manuscript "Abrupt and spontaneous strategy switches emerge in simple regularised neural networks" for consideration at PLOS Computational Biology. 

As with all papers reviewed by the journal, your manuscript was reviewed by members of the editorial board and by two independent reviewers. In light of the reviews (below this email), we would like to invite the resubmission of a revised version that takes into account the reviewers' comments. As you can see from the reviews, both reviewers were very positive about the manuscript, but also had several comments regarding the modelling choices and the human experiments. We would like to see all of the raised points to be fully addressed in a revised version.

We cannot make any decision about publication until we have seen the revised manuscript and your response to the reviewers' comments. Your revised manuscript is also likely to be sent to reviewers for further evaluation.

Sincerely,

Tobias U Hauser, PhD

Academic Editor

PLOS Computational Biology

Daniele Marinazzo

Section Editor

PLOS Computational Biology

Reviewer's Responses to Questions

**Comments to the Authors:**

Reviewer #1: In this interesting manuscript, Löwe et al. conduct well-designed neural network experiments and show how regulated, gated, simple networks can display behaviors similar to human problem-solving insights. There’s a lot to like about this paper: the computational experiments are precise and are very well explained. There are also plenty of control experiments and these findings could be really useful for researchers studying problem-solving, creativity, and insight.

To me, the neural network experiments are very robust and I don't have any major concerns with them. I especially enjoyed the careful control experiments conducted by the authors (particularly the hidden layer experiment in the SI, as this was a major question I had when I had begun reading the article)

However, for the human study, I have one concern which I would like the authors to address. The authors excluded almost half the participants (96 participants) from their analysis, citing insufficient accuracy levels after the motion phase. This is a very high number of subjects to exclude -- why did the authors choose the 85% threshold as the exclusion criteria?

Lastly, it would also be great to see how robust the results are when you change the threshold -- does the pattern of results still hold with different thresholds, e.g., 60% or 70%?

Reviewer #2: Lowe and colleagues investigate whether insight-like behaviours can occur in simple neural networks trained with gradual gradient descent (SGD). They compare a minimal artificial neural network (ANN) toy model and human participants' behaviour in a simple behavioural task. A key property of the task is that a 'hidden' regularity is introduced during learning, which, if noticed, is easily exploitable for solving it. The authors identify three key characteristics of insight-like behaviour: delay, suddenness and selectivity, and show that these are reproducible in their minimal model. They provide a mechanistic understanding of how these characteristics arise in the NN and point out a gating mechanism, noise and regularisation as key ingredients.

The paper is well-organised and I found it easy to read. I believe it is indeed counterintuitive that NNs trained with gradual gradient descent would produce these sudden shifts in behaviour and that therefore a well-understood minimal model has a place in the literature. I think the behavioural experiment, while simple, successfully highlights the essence of the problem and I don't see any major issues with its analysis and interpretation (minor issues listed at the end).

My main critique is regarding architectural choices and their motivation in the minimal model. Most of the architecture seems standard for such basic ANNs, except for the gating mechanism. However, the motivation for this term I believe is not correspondingly spelled out. There are allusions to 'attention' and multiplicative attention is often part of complex architectures such as LSTMs or transformers, but not perceptrons. This seems to me to slightly undercut the combination of "naturally" and "simple" in the main conclusion that "insight-like behaviour can arise naturally from gradual learning in simple neural networks", at least in the sense that it would emerge in commonly used connectionist models. At the very least, the reader would want to know if the point is that attention/gating in more complex architectures is hypothesised to be the key component that enables them to (potentially) exhibit sudden insights; or alternatively is this term perhaps a simplified representation for an emergent gating mechanism in larger, but still relatively basic perceptrons?

The latter possibility (emergent) seems especially relevant as sudden shifts have been shown to emerge in NNs in previous research ('rapid developmental transitions', Saxe, McClelland & Ganguli, 2018). My understanding is that the key ingredient there is the depth of the network. This citation makes an appearance in the intro, but I would advocate for also discussing how the mechanism identified here relates to the one there there, e.g., could the multiplicative gating units be seen as functionally equivalent to a hidden layer that is doing a kind of emergent gating?

Similarly, it was not clear to me why the regularisation term only applies to the gating weights $g_i$ and not the others ($w_i$)?

In line 76: "Technically speaking, regularisation refers to adding a penalty term to the error function that prevents coefficients from reaching large values, and which thereby leads to suppression of input features [41]" I don't think this is accurate or consistent with the perspective of [41], specifically, it muddles the distinction between goal (regularisation) and mechanism (L1 and L2 penalty). Regularisation in general is a method that is intended to prevent overfitting and help with generalisation, a penalty term for large weights being one of the simplest such approaches (but there is also e.g. early stopping, data augmentation and weight sharing, all discussed in section 5.5 in the cited book [41]). In addition to the mentioned sentence, regularisation seems to be used interchangeably with the specific mechanism throughout the paper, but it is not at all made clear why that would be warranted. This is especially confusing in the discussion section where it seems like the connections are often made with regularisation in the broad sense. Do the authors hypothesise that regularisation, understood broadly, is a key component in insight-like behaviour (i.e. in a way that also translates to dreaming as is suggested in the discussion [68]), as opposed to the specific mechanism in this simple neural network happening to interact with another mechanism (multiplicative gating) that way? If yes, what supports this in the current results?

While reading the section where the neural network model is defined, I was trying to evaluate the relative magnitudes of noise sources that are introduced (outcome, gradient, perceptual), but ran into difficulties. I couldn't find the value for $\\sigma$, I might have missed where it is defined but would it be possible to mention this around the definition of Eq1.? Then I was also a bit puzzled by the stimulus representation, for example why colour vs motion were modelled as difference in means vs variance? I see how it probably doesn't make a difference in the end but makes the relation between the experiment and the dynamics of the NN more abstract. In general, I think it would be helpful to clarify the modelling choices for the noise sources a bit, preferably not far from Eq 1-4.

In summary, I think this is an interesting paper, however it focuses on a restricted toy model. On the one hand, this it allows for a clear, mechanistic explanation of the emergence of insight-like behaviour. On the other hand, I feel the motivation for some of the modelling choices are not made sufficiently clear, similarly to the question of how the authors propose this mechanism generalises to non-toy networks on non-toy problems. It would seem useful to me to at least gesture towards what kind of research the understanding achieved in this paper would imply regarding more realistic scale networks and be specifically on what level of abstraction the observations are thought to transferable.

Minor issues and questions:

- Figure 1E, y axis is only made clear in main text, would be good to point it out either on figure or in caption

- Fig 1 caption says "classification of insights versus no insight agreed to 79.6% with verbal insight reports" then in the main text "Of the 49 participants 210 classified as insight subjects 39 (79.6%) also self-reported to have used colour to make 211 correct choices". This doesn't seem to be what I'd understand as agreement between two classifiers, I'd assume one would also want to take into account how much the responses of non-insight subjects (according to self-report) agree with the behavioural measure, maybe this is just unclear phrasing?

- line 271, "in and off"->in and of

**Have the authors made all data and (if applicable) computational code underlying the findings in their manuscript fully available?**

Reviewer #1: **No: **The authors say that all data and code will be made fully available upon publication of this article.

Reviewer #2: Yes

PLOS authors have the option to publish the peer review history of their article (what does this mean?). If published, this will include your full peer review and any attached files.

Reviewer #1: No

Reviewer #2: No
---

## [Decision Letter · Decision Letter 1]

23 Sep 2024

Dear Schuck,

We are pleased to inform you that your manuscript 'Abrupt and spontaneous strategy switches emerge in simple regularised neural networks' has been provisionally accepted for publication in PLOS Computational Biology.

Best regards,

Tobias U Hauser, PhD

Academic Editor

PLOS Computational Biology

Daniele Marinazzo

Section Editor

PLOS Computational Biology

Reviewer's Responses to Questions

**Comments to the Authors:**

Reviewer #1: I thank the authors for thoughtfully engaging with my comment. The authors have addressed my concerns in the revised manuscript, and I am happy to recommend acceptance.

Reviewer #2: I appreciate the authors’ thorough responses to the feedback. I don't have any remaining concerns with the revised manuscript.

**Have the authors made all data and (if applicable) computational code underlying the findings in their manuscript fully available?**

Reviewer #1: Yes

Reviewer #2: Yes

PLOS authors have the option to publish the peer review history of their article (what does this mean?). If published, this will include your full peer review and any attached files.

Reviewer #1: No

Reviewer #2: No

---

## [Editor Report · Acceptance letter]

15 Oct 2024

PCOMPBIOL-D-24-00392R1 

Abrupt and spontaneous strategy switches emerge in simple regularised neural networks

Dear Dr Schuck,

I am pleased to inform you that your manuscript has been formally accepted for publication in PLOS Computational Biology. Your manuscript is now with our production department and you will be notified of the publication date in due course.

With kind regards,

Dorothy Lannert
